# Tree Health Condition in Urban Green Areas Assessed through Crown Indicators and Vegetation Indices

Luis Manuel Morales-Gallegos [1], Tomás Martínez-Trinidad [1,*], Patricia Hernández-de la Rosa [1], Armando Gómez-Guerrero [1], Dionicio Alvarado-Rosales [2] and Luz de Lourdes Saavedra-Romero [2]

1. Posgrado en Ciencias Forestales, Colegio de Postgraduados, Texcoco 56230, Mexico; luis.morales@colpos.mx (L.M.M.-G.); pathr@colpos.mx (P.H.-d.l.R.); agomezg@colpos.mx (A.G.-G.)
2. Posgrado en Fitosanidad, Colegio de Postgraduados, Texcoco 56230, Mexico; dionicioyganoderma@gmail.com (D.A.-R.); saavedra.luz@colpos.mx (L.d.L.S.-R.)
* Correspondence: tomtz@colpos.mx; Tel.: +52-595-952-0246

**Abstract:** The urban environment induces stress on trees and its impact can be identified by observing the condition of the crown. The aim of this study is to correlate the variables of crown density (Cdn), crown transparency (Ctr) and dieback (Cdie) with the following vegetation indices: the normalized difference vegetation index (NDVI), enhanced vegetation index (EVI), blue-normalized difference vegetation index (BNDVI), green-normalized difference vegetation index (GNDVI), green–red vegetation index (GRVI) and red–green–blue vegetation index (RGBVI) of the crowns of trees located in urban green areas, as well as chlorophyll fluorescence (Fv/Fm) as an indirect indicator of the overall tree health condition. A total of 549 trees were evaluated, represented in 24 families, 36 genera and 53 species; the variables had average values of 67.96% for Cdn, 35.19% for Ctr and 1% for Cdie. Correlations were found between Fv/Fm, NDVI and BNDVI. NDVI and BNDVI correlated with variables such as Cdn and Ctr, mainly in species such as *Ligustrum lucidum*, *Jacaranda mimosifolia* and *Fraxinus uhdei*. Therefore, it is possible to evaluate the tree health condition of trees in urban green environments through the identification of unfavorable conditions at the crown level by using vegetation indices for some of the species studied.

**Keywords:** urban trees; vegetation indices; chlorophyll fluorescence; forest health; crown density

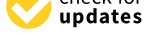



## 1. Introduction

Population growth leads to an accelerated change from a natural environment to an urban landscape; therefore, it is advisable to conserve and increase vegetation through the creation of a greater number of urban green areas (UGAs) [1–3]. The most important vegetation in UGAs are trees, as they provide a wide range of environmental, ecological and social services [4–6]. However, urban trees face stressful conditions due to factors such as the heat island effect, soil compaction, limited growth space for roots, vandalism, inadequate management practices, and water and nutrient deficiency, among others [7–10]. The impact of stress on urban trees can be identified by observing the condition of their crowns. An alteration in their morphological characteristics negatively affects their vitality and general health condition, and also has an impact on the provision of services to the urban environment [6,11–15].

Tree crown assessment is used as an indicator of health condition in forest species, with some variables being crown density, crown transparency and dieback [11,16,17]. Recently, these indicators have been adapted and used to estimate the health condition of urban trees; however, obtaining this information involves in situ data collection by at least two people [4,6,17–19], which entails a considerable expenditure of time and is complicated when access to the terrain is restricted or dangerous; currently, this method is frequently used to obtain information on tree health condition in both forest and urban areas [4,6,10,20]. On the other hand, there are methods to evaluate physiological processes to quantify the

response of trees to stress, one of them being chlorophyll fluorescence (Fv/Fm). This index indicates the photosynthetic efficiency of the system of the leaves; thus, a lower efficiency is related to a lower health condition of the vegetation [21,22].

However, a feasible method for studying tree crowns is the use of unmanned aerial vehicles (UAVs), since being equipped with high-resolution multispectral sensors allows them to obtain precise information from large areas, reducing the time necessary for the analysis of various biophysical parameters compared to traditional methods [9,20,23–27]. Recent vegetation studies have made use of spectral bands to determine vegetation indices (VIs), among other applications; this remote sensing technology allows for the classification and estimation of the health conditions of vegetation in different ecosystems, as well as in urban areas [28–31]. Among the vegetation indices used in research are the normalized difference vegetation index (NDVI), the enhanced vegetation index (EVI), the green-normalized difference vegetation index (GNDVI), the blue-normalized difference vegetation index (BNDVI), the red–green–blue vegetation index (RGBVI) and the green–red vegetation index (GRVI) [5,9,20,26,32–35].

Vegetation indices have advantages over other methods, e.g., NDVI has a better correlation with tree canopy cover than with other ground-level vegetation covers, and high NDVI values indicate healthy vegetation conditions [2,34]. High GNDVI values effectively represent chlorophyll properties, while BNDVI allows the spatial distribution of chlorophyll to be analyzed [26,32,36]. These characteristics allow for analysis of different options in determining the condition of the trees. Additionally, an advantage of this method is that it is objective and repeatable, not observer-dependent like previous subjective measures of tree health. Therefore, the aim of this study was to determine the degree of correlation between the absolute variables of crown density, crown transparency and dieback of trees located in urban green areas with the vegetation indices NDVI, EVI, BNDVI, GNDVI, GRVI and RGBVI, as well as chlorophyll fluorescence, with the purpose of identifying more efficient predictors of tree health condition.

## 2. Materials and Methods

### 2.1. Study Area

The study areas were green areas (UGAs) of the city of Texcoco de Mora (19°30′52.30″ N and 98°52′57.73″ W) in the State of Mexico, Mexico (Figure 1), that concentrate a large number and variety of tree species that allow an overview of the conditions of the trees in these spaces. The city is very flat and the main land cover types in the area include buildings, roads, trees, shrubs and grass. It has an elevation of 2240 m, a temperate climate with an average temperature of 15.9 °C and a mean annual rainfall of 686 mm. The most representative soil has Vertisol-type characteristics; however, the soil has been considerably altered by anthropogenic activities [37].

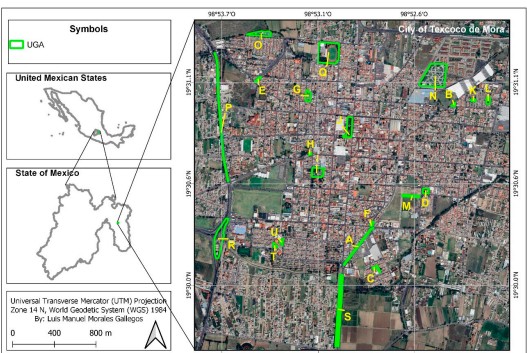

**Figure 1.** Location of green areas in the city of Texcoco de Mora. A–U: Urban green area ID. Google Earth V 7.3.6.9345 (1 January 2021). Texcoco, Mexico. 19°30′52.30″ N, 98°52′57.73″ W, eye alt 20,866 feet. Maxar Technologies 2023. https://www.google.com/intl/es-419/earth/ (accessed on 6 February 2023).

Google Earth images from the year 2021 were used to locate, delimit and dimension 21 UGAs (Table 1) administered by the municipality of Texcoco [38,39]. The UGAs have mainly arboreal vegetation; however, there are also shrub and herbaceous species. A common management practice in the tree crowns is pruning, normally conducted for aesthetic purposes and to avoid contact with nearby infrastructure.

**Table 1.** Urban green areas in Texcoco de Mora, State of Mexico, Mexico.

| ID | Name | Perimeter (m) | Area (m$^2$) | Trees |
|----|------|---------------|--------------|-------|
| A | Boulevard Jiménez Cantú | 940.53 | 4687.21 | 20 |
| B | Valle de Santa Cruz 2 | 141.72 | 848.98 | 14 |
| C | Jardín San Martín | 259.84 | 1379.92 | 18 |
| D | Parque Niños Héroes | 205.03 | 2632.24 | 26 |
| E | Parque las Américas | 174.75 | 1123.34 | 19 |
| F | Parque del Ahuehuete | 132.96 | 876.11 | 16 |
| G | Parque Heberto Castillo | 304.93 | 4167.78 | 24 |
| H | Parque Arteaga | 92.05 | 435.07 | 9 |
| I | Parque de la Tercera Edad | 406.05 | 9478.5 | 49 |
| J | Jardín Municipal | 549.66 | 9765.82 | 44 |
| K | Valle de Santa Cruz 3 | 118.35 | 717.83 | 9 |
| L | Valle de Santa Cruz 1 | 192.78 | 2128.93 | 24 |
| M | Parque Municipal | 366.93 | 2694.09 | 21 |
| N | Alameda Texcoco | 849.57 | 43,898.99 | 71 |
| O | Parque Xolache | 517.07 | 7436.39 | 17 |
| P | Camellón Lechería | 2505.49 | 7554.33 | 19 |
| Q | Deportivo Silverio Pérez | 765.46 | 37,159.45 | 56 |
| R | Parque Bicentenario | 859.26 | 21,397.46 | 23 |
| S | Boulevard Chapingo | 2801.45 | 16,347.42 | 42 |
| T | Las vegas 1 | 151.56 | 1105.62 | 17 |
| U | Las vegas 2 | 167.91 | 1173.21 | 11 |

### 2.2. Collection of Tree Information

A database from a previous study was used in the 21 UGAs of Texcoco, where the trees were surveyed; a representative sample of these was taken to estimate the media of the variables, implementing a simple random sampling for each UGA with a reliability of 95%, using Equation (1) [40]. From the sampled trees, total height (Th) was evaluated with a Haglöf ECII D® electronic clinometer, diameter at breast height (Dbh) was measured with a diameter tape (Forestry Suppliers Inc.® Jackson, MS, USA) and in cases where the trunk was found bifurcated below 1.3 m in height, each trunk was considered as an individual tree [41–43]. Species were identified through botanical keys and field guides; when a full identification was not possible by these means, a botanical collection of the specimen was made and taken to the Hortorio Herbarium (CHAPA) of the Colegio de Postgraduados for identification. Only dominant species with a Dbh ≥ 10 cm (approx. 4 in) were considered, because they provide more spectral information for further analysis; this criterion resulted in a minimum tree height of 1.5 m. Finally, the recording of the information was carried out in a vegetative growth period (August and September 2021) [6,26,33,38,39]

$$n = \frac{N \, \sigma^2 \, Z^2}{(N - 1) \, e^2 + \sigma^2 \, Z^2} \tag{1}$$

where:

$n$ = sample size
$N$ = population size
$\sigma$ = 0.5
$Z$ = confidence level 1.96 (95% confidence)
$e$ = acceptable error limit (5% of sample mean).

### 2.3. Evaluation of Crown Variables

The following variables were assessed: crown density (Cdn), which is a biomass index that includes foliage, branches and reproductive structures; crown transparency (Ctr), which estimates the amount of light passing through the live crown; and dieback (Mrg), which indicates the extent of leafless branches at the periphery of the crown, generally increasing from the top to the bottom of the tree [44]. For this, the field assessment variables designed by the Forest Inventory and Analysis (FIA) program of the United States [17,45] were used; this assessment was carried out visually by two people located at a distance proportional to the height of the tree to be measured. When they had a different opinion, a consensus was reached under an average value. Despite the fact that variables were recorded with values in percentage increments of 5% on a scale from 0 to 100 [4,6], in order to avoid a rounding down effect or ranges with few results, data were analyzed in increments of 10%.

### 2.4. UAV Multispectral Images

We used a DJI PHANTOM 4$^©$ UAV equipped with six 1/2.9″ CMOS sensors, one RGB sensor and five monochrome sensors with the ability to capture images in the blue (B: 450 nm $\pm$ 16 nm), green (G: 560 nm $\pm$ 16 nm), red (R: 650 $\pm$ 16 nm), red-edge (RE: 730 $\pm$ 16 nm) and near-infrared (NIR: 840 $\pm$ 26 nm) spectra, which are widely used in the study of both forest and urban vegetation [9,26]. The UAV has a resolution of 9.52 cm/pixel at an altitude of 180 m, with a vertical and lateral overlap of 80 and 60%, respectively, and a maximum operating area of 0.63 km$^2$; it has a field of view of 62.7° (HFOV) and a weight of 1487 g. The DJI GS Pro$^©$ was used to establish the UAV flight plan; photography was performed at a flight altitude of 60 m, which obtained images with a resolution of 0.3 m. Radiometric calibration was performed through the Calibrated Reflectance Panel (CRP) and the camera's incident light sensor. The UAV has an integrated GPS/GLONASS system, allowing for faster and more accurate satellite acquisition during flights, as well as eight pre-set checkpoints on each UGA to ensure correct image positioning. Finally, ortho-mosaics (.tiff) of each UGA were created using Pix4D Mapper software v. 4.8.4 (Lausanne, Switzerland) [26,27,46]. Flights were conducted in late August and early September 2021 in the summer season, a time of year in which the vegetation is in a state of vegetative growth due to constant rainfall and temperatures above the annual average, at times from 11:00 to 13:00 h, with winds < 5 km/h to avoid distortions due to movement.

### 2.5. Vegetation Indices

Multispectral images (B, G, R and NIR) from the UAV were used to calculate the NDVI, EVI, GNDVI, BNDVI, RGBVI and GRVI indices (Table 2) [26]. Given the precision of the UAV images (0.3 m), it was possible to delimit the crowns of the trees defined in the previous sampling manually (digitization) [27]; this information was transformed from raster to vector format for processing and analyzed through the QGIS version 3.28.4 Firenze geographic information system (GIS) [9,20,24]. This provided greater precision in choosing the pixels of each tree's crown and excluding background information, such as that of another type of vegetation (shrub or herbaceous) or of nearby trees not belonging to the study, to later obtain the average values of the pixels that make up the image of each tree [2,9,26] (Figure 2). On the other hand, the use of different vegetation indices also allows us to detect the alterations caused by tree flowering, which can vary the green tree color of the foliage, something that is difficult to control due to the number of tree species under study.

Also, the coefficient of variation (Equation (2)) of each vegetation index per crown was determined; this allows for a better comparison between indices, and it is a widely used measure in vegetation research that reflects the discrete degree of the data. The CVs were

categorized into High and Low using the percentile ranges of >3rd percentile and <3rd percentile, respectively [35,47,48].

$$CV = \frac{\sigma}{\bar{x}} * 100 \tag{2}$$

where:

$\sigma$ = Standard deviation
$\bar{x}$ = Arithmetic mean

**Table 2.** Equations used to determine vegetation indices.

| Formulas | Where |
|---|---|
| $NDVI = \frac{Nir-Red}{Nir+Red}$ | $NDVI$ = Normalized difference vegetation index<br>$Nir$ = Near infrared<br>$Red$ = Red band |
| $EVI = \frac{2.5*(Nir-Red)}{Nir+Red+1}$ | $EVI$ = Enhanced Vegetation Index<br>$Nir$ = Near infrared<br>$Red$ = Red band |
| $GNDVI = \frac{Nir-Green}{Nir+Green}$ | $GNDVI$ = Green-normalized difference vegetation index<br>$Nir$ = Near infrared<br>$Green$ = Green band |
| $BNDVI = \frac{Nir-Blue}{Nir+Blue}$ | $BNDVI$ = Blue-normalized difference vegetation index<br>$Nir$ = Near infrared<br>$Blue$ = Blue band |
| $GRVI = \frac{Green-Red}{Green+Red}$ | $GRVI$ = Green–red vegetation index<br>$Red$ = Red band<br>$Green$ = Green band |
| $RGBVI = \frac{Green^2-(Red*Blue)}{Green^2+(Red*Blue)}$ | $RGBVI$ = Red–green–blue vegetation index<br>$Red$ = Red band<br>$Green$ = Green band<br>$Blue$ = Blue band |

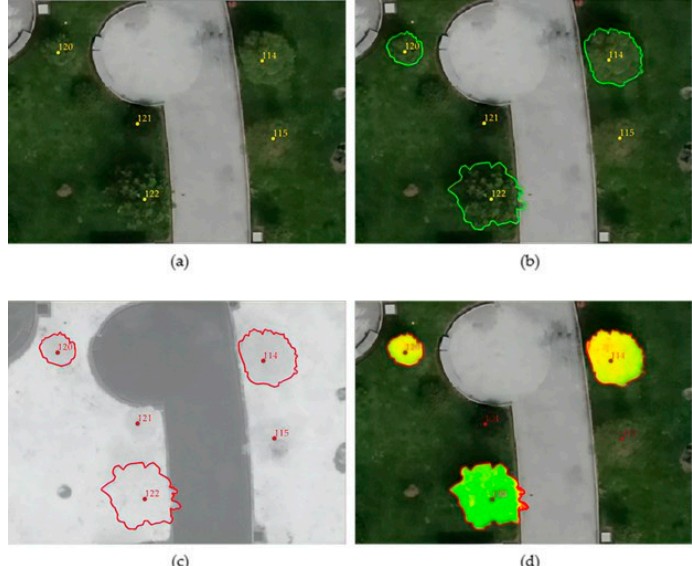

**Figure 2.** Example applied to the six evaluated indices and 21 UGAs studied. (**a**) Location of the tree species under study prior to sampling. (**b**) Digitization of crowns. (**c**) Calculation of the vegetation index. (**d**) Extraction of pixel values of each tree's crown index. Small numbers inside the tree crown shape correspond to the tree identifier number in the field.

### 2.6. Health Condition

Chlorophyll fluorescence (Fv/Fm) was assessed in the sampled tree species as an indirect measure of the overall tree health through their physiological stress condition [49]. A portable Pocket PEA fluorimeter (Hansatech Instruments Ltd., King's Lynn, UK) was used, with a detection parameter of 1 s and light emission at a wavelength of 650 nm with an intensity of 3500 $\mu$mol m$^{-2}$ s$^{-1}$ [22]. The measurements were made by adapting to the dark and using the fluorimeter clips for 10 min on a total of 5 leaves chosen at random around each tree's crown [50]. All activities were carried out between August and September 2021, and are integrated in the methodological diagram shown in Figure 3.

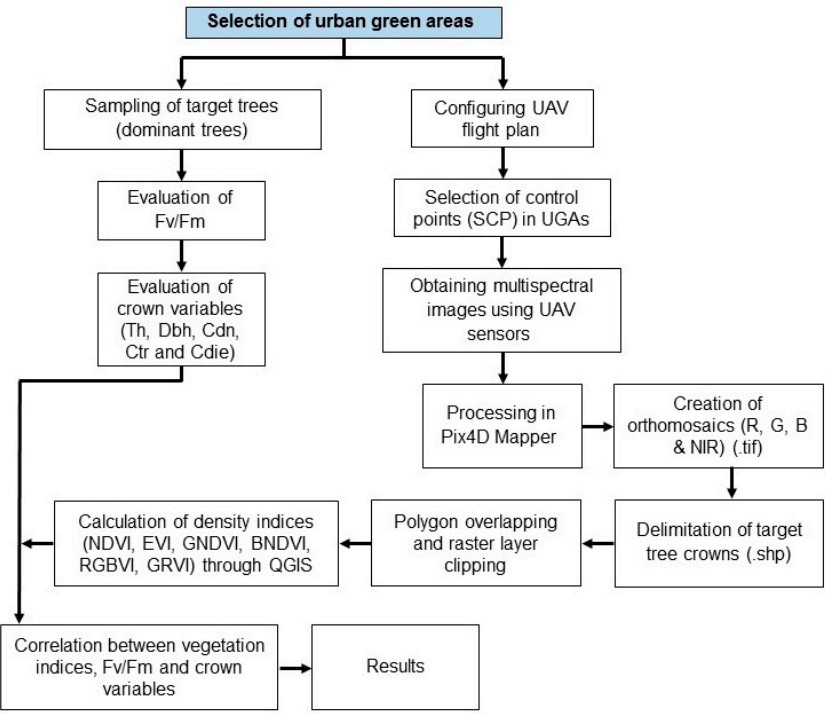

**Figure 3.** Methodological diagram of the study.

### 2.7. Data Analysis

A database was created in Microsoft Excel$^{@}$ using dynamic tables to organize the information, which facilitated its analysis [42]. Correlations (Pearson) were carried out between the variables Fv/Fm, NDVI, EVI, BNDVI, GRVI, RGBVI, Cdn, Ctr, Cdie, Dbh and Th at the species level considering only the 5 most frequent species (species with the best establishment) in each UGA to find associations. In addition, linear regressions were carried out to estimate the values of the dependent variable for unobserved values of the independent variable. The $p$-value $< 0.05$ was selected as the limit of statistical significance. Statistical analyses and histograms were carried out in RStudio software using the base version of R 4.1.3 [51].

### 3. Results

A total of 549 trees were evaluated, represented in 24 families, 36 genera and 53 species, of which Cupressaceae, Rosaceae, Fabaceae, Oleaceae, Bignoniaceae and Pinaceae were the most frequent, representing more than 50% of the evaluated species. The diameter classes indicated that 71.95% of the trees' diameters were found to be within the 10–30 cm range (Figure 4A). Regarding heights, the first four categories represent 66.67% of the evaluated medium-sized trees (5.5–7.5 m) (Figure 4B).

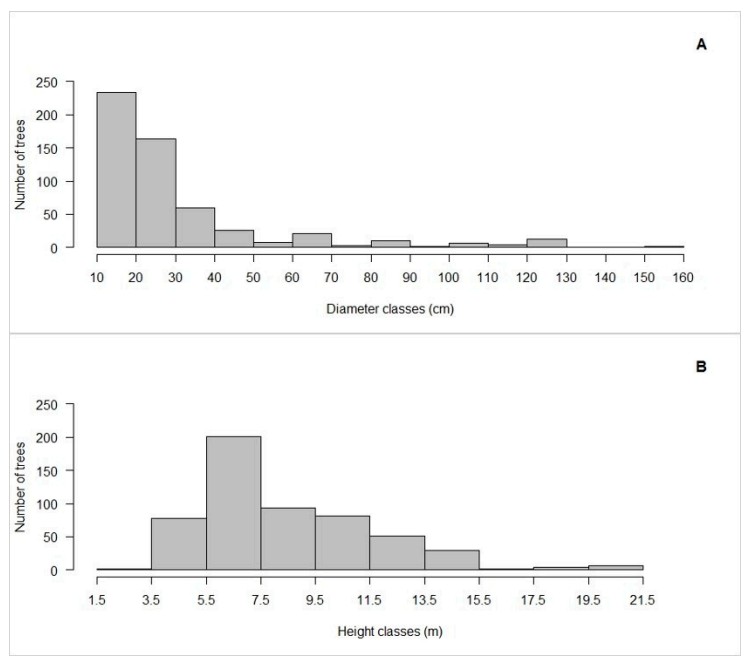

**Figure 4.** Diameter (**A**) and height (**B**) categories in 549 trees located in UGAs in the city of Texcoco de Mora.

On the other hand, tree density was estimated at 50 individuals on average per UGA, giving a density of 130 trees per hectare. At the species level, Dbh and Th presented significant statistical differences ($p < 0.05$), finding that *Schinus molle* L. presented the largest diameter ($52.65 \pm 6.65$ cm) and *Casuarina equisetifolia* L. the greatest height ($9.8 \pm 0.79$ m). Conversely, *Cupressus sempervirens* L. had the smallest diameter ($13.54 \pm 0.76$ cm) and *Ficus benjamina* L. the lowest height ($4.53 \pm 0.23$ m).

The tree crown density values were located mostly within a range of 40% to 100%, with a tendency towards a mostly dense crown (Figure 5A); Ctr showed a main range from 0% to 40% (Figure 5B), indicating that most of the trees have low crown transparency values. Finally, Cdie showed a very low percentage, that is, little damage due to this degenerative condition (Figure 5C).

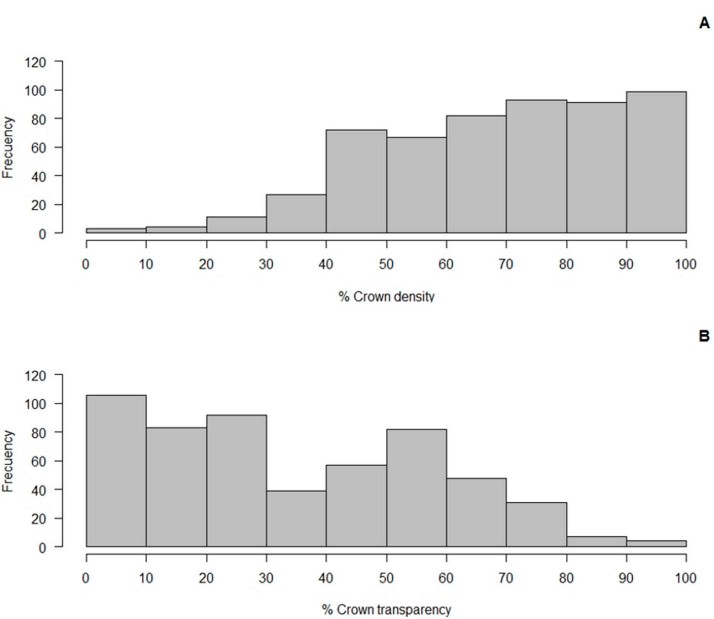

**Figure 5.** *Cont.*

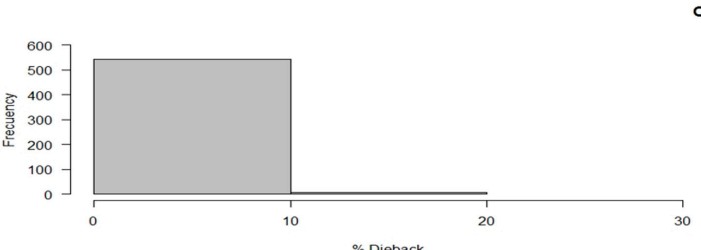

**Figure 5.** Histograms of frequencies for the variables of crown density (Cdn) (**A**), crown transparency (Ctr) (**B**) and dieback (Cdie) (**C**) in UGAs in the city of Texcoco de Mora.

*3.1. Correlations between the Variables Studied*

The correlations studied between vegetation indices, crown variables and chlorophyll fluorescence indicated some associations (Figure 6).

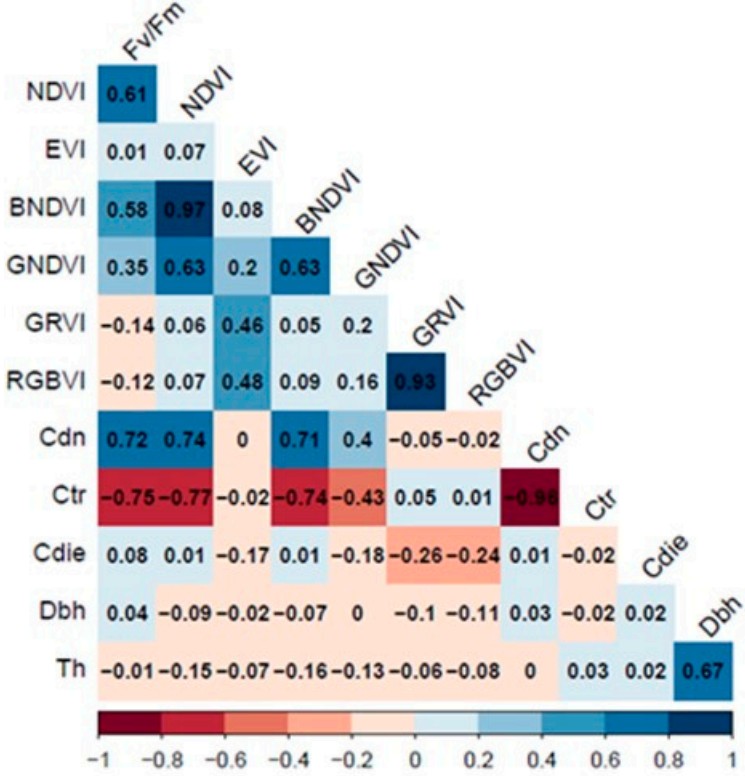

**Figure 6.** Pearson correlation matrix [®] between vegetation indices, chlorophyll fluorescence and tree variables ($p < 0.05$). Fv/Fm: chlorophyll fluorescence, NDVI: Normalized difference vegetation index, EVI: Enhanced vegetation index, BNDVI: Blue-normalized difference vegetation index, GNDVI: Green-normalized difference vegetation index, GRVI: Green–red vegetation index, RGBVI: Red–green–blue vegetation index, Cdn: Crown density, Ctr: Crown transparency, Cdie: Crown dieback, Dbh: diameter at breast height and Th: Total height.

Chlorophyll fluorescence (Fv/Fm) was found to correlate positively with NDVI ($r = 0.61$, $p < 0.05$), as well as with Cdn ($r = 0.72$, $p < 0.05$), and negatively with Ctr ($r = -0.75$, $p < 0.05$), and to a lesser degree correlated with BNDVI ($r = 0.58$, $p < 0.05$). NDVI also correlated positively with Cdn ($r = 0.74$, $p < 0.05$) and negatively with Ctr ($r = -0.77$, $p < 0.05$). On the other hand, BNDVI presented a correlation similar to NDVI with the variables Cdn ($r = 0.71$, $p < 0.05$) and Ctr ($r = -0.74$, $p < 0.05$). Other correlations found

were between Cdn and Ctr (r = −0.96, *p* < 0.05) and diameter and tree height (r = 0.67, *p* < 0.05) (Figure 6).

Some linear regressions indicate that some independent variables have a significant effect in explaining the dependent variable (Figure 7), mainly between the NDVI, BNDVI, Cdn and Ctr of the most frequent studied species.

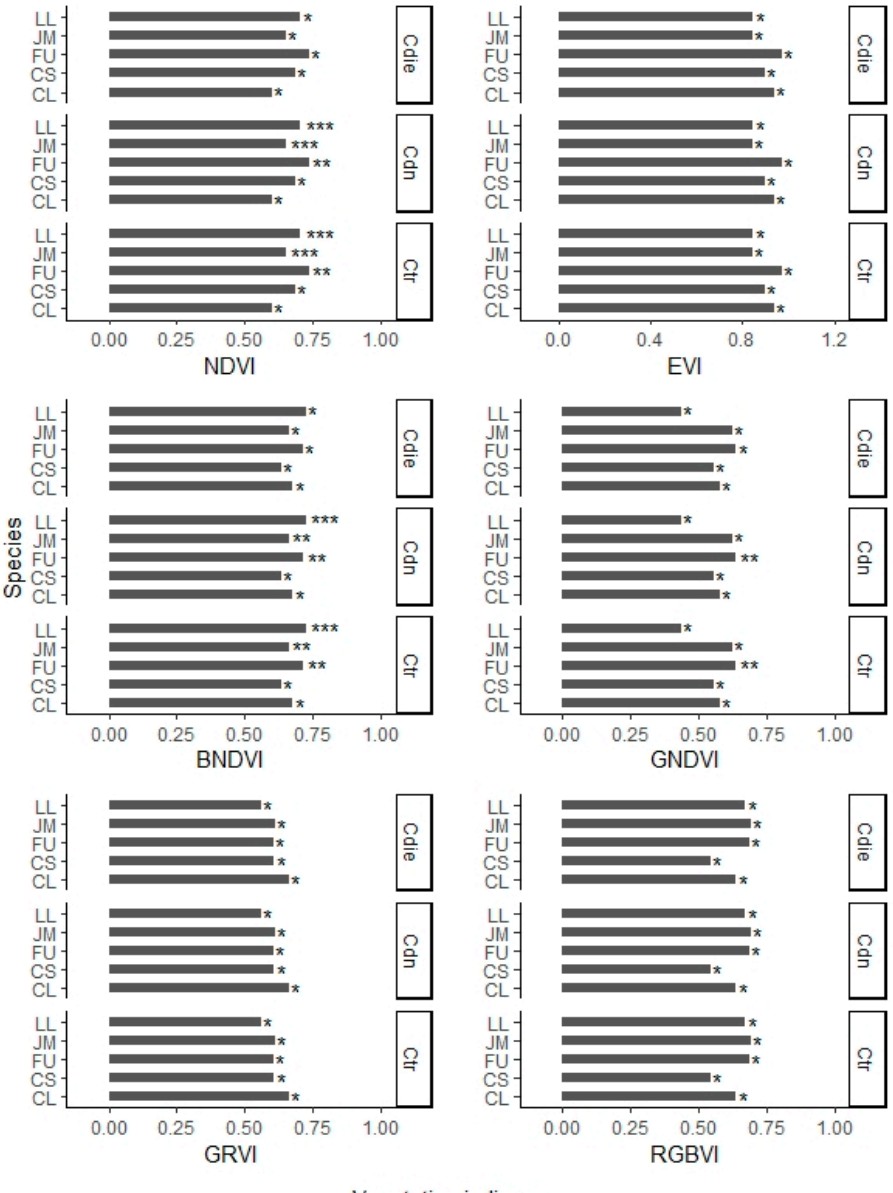

**Figure 7.** Linear regressions between the five most frequent tree species in the study area, crown indicators and vegetation indices evaluated. Adjusted R2: * < 0.5; ** = 0.5; *** > 0.5. LL: *Ligustrum lucidum* W. T. Aiton; JM: *Jacaranda mimosifolia* D. Don; FU: *Fraxinus uhdei* (Wenz.) Lingelsh.; CS: *Cupressus sempervirens* L.; CL: *Cupressus lusitanica* Mill.; NDVI: Normalized difference vegetation index; EVI: Enhanced vegetation index; BNDVI: Blue-normalized difference vegetation index; GNDVI: Green-normalized difference vegetation index; GRVI: Green–red vegetation index; RGBVI: Red–green–blue vegetation index; Cdn: Crown density; Ctr: Crown transparency; and Cdie: Crown dieback.

### 3.2. Coefficients of Variation (CVs) of Vegetation Indices

The analysis of the CVs of the vegetation indices indicated a greater variation in the trees with low index values (<3rd quartile); this was for all the indices evaluated. On the contrary, trees with high index values had a lower dispersion of data.

## 4. Discussion

The smaller diameters and heights of the evaluated trees (Figure 4) indicate that the species may be mostly young (Figure 4A) [20]. It was also found that they are short (Figure 4B), which may be due to pruning activities on the tree crowns. This result is contrary to what was found, for example, in the city of Montemorelos, Nuevo León, Mexico, where the urban trees are larger [52], which may indicate that cultural practices are an important factor affecting the structure and composition of tree vegetation in urban areas. On the other hand, older trees would be expected to have crowns with high density (Cdn) values and low percentages of crown transparency (Ctr) and dieback (Cdie); however, maintenance actions carried out on the crowns (formation pruning) alter their shape and condition, diminishing the vitality of the trees and, in extreme cases, leading to their death [6,20].

In general, trees are considered healthy when they present values of Cdn > 50%, Ctr < 30% and Cdie < 5% [6]. The values of Cdn, Ctr and Cdie found in the trees located in the 21 UGAs of the city of Texcoco have average values of 67.96% in Cdn, 35.19% in Ctr and 1% in Cdie (Figure 5). A high crown density value indicates that the tree has a large number of leaves, which translates into greater photosynthetic capacity and therefore better growth and development [9,14]. In contrast, a low Cdn value translates into little foliage, which can result in physiological stress and greater susceptibility to pest and disease attacks [20,46]. As for Ctr, it showed values above 30% (Figure 5B), indicating that these trees are under stress; however, an annual monitoring of the increase in crown transparency in the trees would help to determine if their growth is compromised, indicating medium-term damage to their reproductive potential and long-term consequences for their survival. Cdn and Ctr can vary by species, age, genotype and evaluation periods. Despite this, at present, these variables are widely used as useful indicators in the evaluation of tree crown condition in both natural and urban environments [6,22]. Regarding Cdie, it presented low frequency within the trees in the 21 UGAs evaluated (Figure 5C). Trees with high Cdie values generally exhibit poor structural conditions, an irregular crown shape and little foliage; therefore, a value higher than 5% in Cdie would indicate that they are not healthy trees [6,14]. The urban environment is a stress factor for trees, with a lack of water being one of the factors present in urban areas which mainly affects variables such as Cdie [4,6].

One way of estimating the vitality of vegetation is based on the amount of chlorophyll present in its leaves, with the use of multispectral images having become a fast and low-cost alternative for estimating chlorophyll content [48]. This work found that NDVI correlated positively with Fv/Fm (Figure 6), given that chlorophyll fluorescence evaluates the photosynthetic activity of the leaves and NDVI is sensitive to chlorophyll. An association was found between these variables, showing that NDVI can help to better evaluate the condition of the tree crown than other types of indices. This is due to the polyfunctionality of NDVI, which achieves good results in different environments; it also serves as a point of comparison with other indices [2,34,53]. On the other hand, a low correlation may be due to several reasons; one that has been studied recently is flowering phenology, which interferes with the spectral bands; that is, the degree and variety of colors present in the flowers alter the "greenness" recorded by the index and may vary according to the time of the year in which it is evaluated [24,36]. BNDVI, however, showed a lower correlation with Fv/Fm (Figure 6), despite the fact that this index helps in the analysis of spatial heterogeneity and chlorophyll distribution; in contrast, a study on bryophytes found a positive correlation between BDNVI and Fv/Fm, which is attributed to the characteristics of the vegetation studied (non-vascular plants), as well as the diversity of species studied [32,54].

Another evaluated vegetation index which is an indicator of the greenness of the tree canopy is the GNDVI; in this case, it did not show a significant correlation with Fv/Fm (Figure 6), possibly due to the same flowering condition mentioned above. However, a study on the species *Coffea Arabica* L. found that some of the indicators related to the leaf content (chlorophyll) of this species were NDVI and GNDVI [48]; it should also be noted that GNDVI has recently been used to estimate the floral proportion in tree crowns located

in natural forests at the pixel level with an accuracy >85% [36]. In this case, it is important to point out that among the tree species that flowered to different degrees during the study were *Bauhinia variegata* L. (pink, purple and white flowering), *Spathodea campanulata* P.Beauv. (orange flowering), *Talipariti tiliaceum* (L.) Fryxell (pink and yellow flowering), *Jacaranda mimosifolia* D Don. (purple flowering) and *Grevillea robusta* A. Cunn. ex R. Br. (yellow and orange flowering), of which only jacaranda was among the most frequent species in the UGAs. Although the correlations between Fv/Fm and the vegetation indices were low, it is also important to note that significant statistical differences were found ($p < 0.05$).

On the other hand, the correlations between Fv/Fm and both Cdn and Ctr (Figure 6) indicate a reference to the health condition of the trees, since it is known that the presence of sparse crowns or those with high percentages of transparency are indicative of stress, which can be evaluated through Fv/Fm [22,50]. Fv/Fm is widely used in crop analysis; however, its use has rapidly spread to other natural and non-natural environments, so it is also used in urban tree stress assessment [32,49]. For example, some studies have detected stress in tree plant species through Fv/Fm under various cultural practices within UGAs, which include management and maintenance activities, particularly crown pruning or transplanting of urban trees [6,22,49]. Finally, NDVI and BNDVI were found to correlate with Cdn and Ctr, this given that they use a similar mathematical relationship [48]; however, the crown variables are better related to NDVI. It is important to emphasize that the proposed method is objective, repeatable and not observer-dependent like previous subjective measures of tree health.

The associations between indices, such as the high correlation found between BNDVI and GNDVI (Figure 6), indicate that the variability not explained by one index can be explained by the other, and, therefore, they complement each other [26,32]; this allows for the development of predictive models [48]. In this sense, the regressions between crown variables and vegetation indices showed associations between NDVI, BNDVI, Cdn and Ctr, which have been described in other studies [9,26,34,55]; one of the most noteworthy correlations is NDVI with tree density, which has $r^2$ values greater than 0.7. This indicates that with these data, it is possible to generate predictive models [28]. With the growth of urban areas and the incorporation of various artificial elements, the generation of predictive vegetation index models with greater accuracy becomes essential. In this sense, a study classified vegetation using NDVI, GNDVI, BNDVI, RGBVI, GRVI and SAVI (Soil Adjusted Vegetation Index) and developed an index that discriminates urban elements such as steel roofs and waterproofing, among others, when these predominate in the images [26].

The CVs revealed that there are differences between high and low index values. The results suggest that the evaluation of chlorophyll (green color) is not consistent with low index values, which may be caused by some anthropogenic damage [35,48]. Studies on crops such as coffee (*C. arabica*), in which various plant indices are evaluated in diseased and healthy leaves, indicate that a high CV may represent a non-uniform distribution of chlorophyll for diseased leaves, and this may be due to various factors that cause the degradation of the pigment (chlorophyll), such as trauma, chemicals, infectious agents and senescence stages, among others [28,48].

## 5. Conclusions

It is possible to evaluate tree urban health condition by using vegetation indices, such as NDVI, BNDVI and GNDVI, using drones. Variables such as chlorophyll fluorescence, crown density and transparency were significantly related to the vegetation indices using the near infrared (NIR) band, so it is feasible to use them and include them in the design of new indices to assess the condition of urban tree vegetation. Generating predictive health condition models is possible by considering positive correlations between vegetation indices with low coefficients of variation. The identification of unfavorable conditions at tree-crown level through the use of vegetation indices allows for timely actions to be taken related to their management, making this a pioneering study on urban trees in

Mexico. Although the correlations between chlorophyll fluorescence and vegetation indices were low, what is important is the acceptable statistical significance ($p < 0.05$). Finally, the use of other types of indices such as the greenness index (GR) or the atmospheric resistant vegetation index (ARVI) are recommended to reduce the effects of flowering in urban trees and to improve the assessment of their health condition with the use of multispectral images.

**Author Contributions:** Conceptualization, L.M.M.-G., T.M.-T. and L.d.L.S.-R.; methodology, L.M.M.-G., T.M.-T., P.H.-d.l.R., D.A.-R. and A.G.-G.; validation, L.M.M.-G., T.M.-T. and P.H.-d.l.R.; formal analysis, L.M.M.-G., T.M.-T., D.A.-R. and P.H.-d.l.R.; investigation, L.M.M.-G., T.M.-T., D.A.-R. and L.d.L.S.-R.; writing—original draft preparation, L.M.M.-G., T.M.-T. and A.G.-G.; writing—review and editing, L.M.M.-G., T.M.-T., D.A.-R., P.H.-d.l.R. and L.d.L.S.-R.; visualization, L.M.M.-G. and T.M.-T.; supervision, T.M.-T., D.A.-R. and P.H.-d.l.R.; project administration, L.M.M.-G. and T.M.-T.; funding acquisition, L.M.M.-G. and T.M.-T. All authors have read and agreed to the published version of the manuscript.

**Funding:** This research was partially supported by the *Colegio de Postgraduados*' Structural and Functional Improvement of Forest Ecosystems' Knowledge Generation and Application Line.

**Data Availability Statement:** Data can be obtained from the authors upon reasonable request.

**Acknowledgments:** The authors thank the National Council of Science and Technology (CONACYT) in terms of the scholarship granted to the first author for graduate studies and the academic editor, as well as Javier Suárez Espinosa for his valuable contribution in the review of the statistical analyses and comments to improve this work.

**Conflicts of Interest:** The authors declare no conflict of interest.

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
