# Peer review of "Tree Health Condition in Urban Green Areas Assessed through Crown Indicators and Vegetation Indices"

_forests, doi:10.3390/f14081673_

Round 1
Reviewer 1 Report
This is a well-structured paper with clear methodological and evaluation background. Tree health condition survey has developed dynamically in the past decades. Among contemporary green area and green intensity analysis methods, the NDVI, GNDVI measurements based on satellite-collected databases have been generally used for urban green system analyses and for a primary tree canopy survey. The technical and digital improvements and drone techniques resulted in the development of a much more detailed survey system that can be applied at the micro-level. The new tree health condition survey has many advantages and seems very useful in green areas and especially tree management.
However, the drone-supported micro-scale measurement and survey system may have other advantages too. UAV studies can be more resilient and adaptable when small-scale measurements are needed, or in special weather conditions like urban heat islands, or long heat periods etc.; even the various ecological features of urban green areas could be detected via micro-scaled NDVI surveys. Therefore, the tree health condition survey should open its focus on green system evaluation too. Traditional site analyses and tree surveys proved that the health condition and hence the ecological supply of urban green areas and especially tree canopy are strictly affected by the neighbouring urban areas' functions, the environmental feature and pollution. The overall ecological quality of green areas depends on the physical, and geographical form, and position within the urban fabric. A longitudinal green area, like urban allées or greenways, might be strongly affected and polluted by the neighbouring urban functions, like motorways, roads, heavy traffic and the strong artificial heat island effect; so the urban position can be a challenging feature for the trees and the vegetation in general. Similarly, large urban parks offer different ecological features in the protected inside areas and in the boundaries.
The territorial, green system aspect of the methodology is missing. What is the connection between the urban green areas listed in Table 1 and the selection of 549 trees for the survey? Wonder if all UGA types are represented in the selection - from an ecological point of view, like longitudinal, large urban park small urban open space. Table 1 lists 21 urban green areas but no information is given about the tree selection of these greens. I warmly propose collecting your tree data into a new table, where you give the number and the taxon of the trees enrolled on the survey. Moreover, make a survey of the urban green areas, defining the typical green areas' type on the most important ecological features and characters, like the territorial and formal characteristics and the position of survey trees within the green areas. This urban system analysis could help define possible coherences between tree health conditions and urban fabric.
I am not a native of English and so not enough qualified to professionally assess the quality of English in this paper, still, I find English fine, easy to understand and good fluent with only minor mistakes in spelling. A simple checking could be enough.
Author Response
We thank to the reviewers for the time and effort put into reviewing this manuscript. We have worked to build upon the constructive recommendations to improve the manuscript in this revision.
- To the comment “Therefore, the tree health condition survey should open its focus on green system evaluation too”, it is worth mentioning that the objective of the present study was to evaluate the health condition of urban trees in green areas, although and based on the results obtained, it is possible to apply this type of analysis in the future to the urban green system.
- Regarding the comment “The territorial, green system aspect of the methodology is missing”, the complementary information on the methodology was integrated (Pg. 2, Ln. 78 and 79).
- Regarding the question “What is the connection between the urban green areas listed in Table 1 and the selection of 549 trees for the survey?”, the context is included in Pg. 2, Ln. 76-78.
- In the comment “Wonder if all UGA types are represented in the selection - from an ecological point of view, like longitudinal, large urban park small urban open space”, it should be noted that Table 1 shows the various characteristics of the UGAs that include large and smaller areas (Pg. 3, Ln. 92).
- Reviewer #1 points out that “Table 1 lists 21 urban green areas but no information is given about the tree selection of these greens”, which clarifies that the selection of trees is present in the methodology on Pg. 4 , Ln. 95 and 96.
- To address the comment “I warmly propose collecting your tree data into a new table, where you give the number and the taxon of the trees rolled on the survey”, a column is added in Table 1 named “Trees” corresponding to the number of trees by UGAs (Pg. 3, Ln. 92). Regarding the taxa, the most frequent ones are mentioned in the results section (Pg. 7, Ln. 220-222).
- Finally, to the suggestion of “Moreover, make a survey of the urban green areas, defining the typical green areas' type on the most important ecological features and characters, like the territorial and formal characteristics and the position of survey trees within the green areas. This urban system analysis could help define possible coherences between tree health conditions and urban fabric”, it should be noted that the work is in an initial stage whose objective was rather exploratory, so no other type of analysis was included, however, the results offer the possibility of further investigation under various conditions.
Reviewer 2 Report
This manuscript presents a novel means by which urban plant health can be measured and monitored. The use of definite quantitative values by which trees can be assessed, combined with methods that adequately compare trees, presents a means by which cities can monitor the health of their trees. The low-cost approach indicates that this paper could represent the future of urban tree monitoring.
However, although the authors present a useful method and metric for assessing tree health condition (THC), the paper falls flat by spending too much time comparing the correlations between assessment methods and too little showing how a novel method of THC assessment can benefit a city.
If the authors could compare two UGAs with known different THCs due to pollution, heat islands or soil compaction, and then validate their methodology by showing that their measures confirm known THC, this would be a slam-dunk of a manuscript. Even comparing within-species THC during different seasons using the indicators and indices would be useful. The authors probably have the data to do this.
The methods presented are only useful for comparing trees within a species. The authors do explain this limitation (266-267), but it needs to be a larger component of the study. This is an asset, not a liability. Using the same species in different neighborhoods is an excellent way to compare THC as a factor of neighborhood variables.
In lines 247-253, the authors indicate how THC can be a function of past maintenance, so it is evident they have the data to test hypotheses and indicate the value of their work.
The authors need to explain their methods for controlling for flowering in the methods section, not introduce how it was a problem in the discussion.
Major Revisions
Table 3 is irrelevant. Cities plant different trees at different times and in different neighborhoods. My city uses planetrees in one neighborhood and red maple in another. The neighborhoods are adjacent, but were built 20 years apart, so the planetrees have larger diameters and heights. Showing this in a paper is not novel.
Figure 7 can be made much more readable by replacing it with some examples of the most correlated factors or most interesting correlations. As is, the figure is very dense and filled with irrelevant data.
Figure 8 can be removed without impacting the message of the manuscript.
Minor revisions
Line 151- Extra periods
Line 179- Add edition for R, cite in
Figure 4- No need for line functions or r-squared values here
Figure 5- Cumulative frequency does not matter
Figure 6- Values of 1.00 can be excluded. We assume autocorrelation.
Again, this is an excellent methodology, it just needs to be presented as a means to test hypotheses, not a bunch of indicators that correlate with other indicators. Combining drones with analytical software is seminal, and when AI can be used to assess the data, flyovers can be used to find where THC is low and funds can be properly allocated to save trees!
Author Response
We thank to the reviewers for the time and effort put into reviewing this manuscript. We have worked to build upon the constructive recommendations to improve the manuscript in this revision.
- To the comment on: “However, although the authors present a useful method and metric for assessing tree health condition (THC), the paper falls flat by spending too much time comparing the correlations between assessment methods and too little showing how a novel method of THC assessment can benefit a city", it is indicated that an important point of the work is to know the relationship between the indices, crown variables and tree health, although it is true that in general the health condition of urban trees depends on several factors Both of the green areas and of the geography and climate of the study area, the results found in the present work will allow the analysis to be extrapolated to other conditions of the urban environment, being something that is beyond the objectives of the present study.
- In relation to the suggestions expressed in “If the authors could compare two UGAs with known different THCs due to pollution, heat islands or soil compaction, and then validate their methodology by showing that their measures confirm known THC. Even comparing within-species THC during different seasons using the indicators and indices would be useful. The authors probably have the data to do this. The methods presented are only useful for comparing trees within a species. The authors do explain this limitation (Ln 266-267), but it needs to be a larger component of the study. This is an asset, not a liability. Using the same species in different neighborhoods is an excellent way to compare THC as a factor of neighborhood variables.”, It should be noted that the need to validate the methodology presented in this study is understood, however, in order to meet the stated objectives, it is mentioned that this type of analysis, as indicated by Reviewer #2, is outside the scope of the work.
- To the comment “In lines 247-253, the authors indicate how THC can be a function of past maintenance, so it is evident they have the data to test hypotheses and indicate the value of their work”, the fact of complying with the proposed objectives and scope of this study.
- Regarding the comment “The authors need to explain their methods for controlling for flowering in the methods section, not introduce how it was a problem in the discussion”, information is added in the methodology section (Pg. 5, Ln. 153 -155).
Major revisions
- At the major revision “Table 3 is irrelevant”, Table 3 is removed and supplemented with some values (Pg. 7, Ln. 255 - 257).
- Regarding “Figure 7 can be made much more readable by replacing it with some examples of the most correlated factors or most interesting correlations. As is, the figure is very dense and filled with irrelevant data”, we consider keeping Figure 7 since it compares the vegetation indices with the crown condition variables of the species with the best establishment, which gives an overview of their condition. (Pg. 10, Ln. 337).
- Regarding the comment “Figure 8 can be removed without impacting the message of the manuscript”, the figure and reference are removed (Pg. 10, Ln. 348).
minor revisions
- The suggestion “Line 151- Extra periods”, the required change or suggestion cannot be found.
- To the comment “Line 179- Add edition for R, cite in”, the complementary information is added (Pg. 7, Ln. 218).
- In the minor revision “Figure 4- No need for line functions or r-squared values here”, the information in the graphs is omitted at the suggestion of Reviewer #2 (Pg. 7, Ln. 249).
- In the minor revision “Figure 5- Cumulative frequency does not matter”, the corresponding line is removed (Pg. 8, Ln. 302).
- Regarding the suggestion of “Figure 6- Values of 1.00 can be excluded”, the change to the graph was made (Pg. 9, Ln. 321).